# Prognostic Factors in Patients Diagnosed with Gallbladder Cancer over a Period of 20 Years: A Cohort Study

**DOI:** 10.3390/cancers16172932

**Published:** 2024-08-23

**Authors:** Nima Toussi, Krishna Daida, Michael Moser, Duc Le, Kimberly Hagel, Rani Kanthan, John Shaw, Adnan Zaidi, Haji Chalchal, Shahid Ahmed

**Affiliations:** 1College of Medicine, University of Saskatchewan, Saskatoon, SK S7N4H4, Canada; nima.toussi@usask.ca (N.T.); krd154@usask.ca (K.D.); mam305@usask.ca (M.M.); duc.le@saskcancer.ca (D.L.); kimberly.hagel@saskcancer.ca (K.H.); rani.kanthan@saskhealthauthority.ca (R.K.); john.shaw@usask.ca (J.S.); adnan.zaidi@saskcancer.ca (A.Z.); haji.chalchal@saskcancer.ca (H.C.); 2Department of Surgery, University of Saskatchewan, Saskatoon, SK S7N0W8, Canada; 3Department of Oncology, University of Saskatchewan, Saskatoon, SK S7N4H4, Canada; 4Saskatoon Cancer Center, Saskatoon, SK S7N4H4, Canada; 5Allan Blair Cancer Center, Regina, SK S4T7T1, Canada; 6Canada Department of Pathology, University of Saskatchewan, Saskatoon, SK S7N0W8, Canada

**Keywords:** gallbladder cancer, rural residence, survival, surgery, chemotherapy, neutrophil–lymphocyte ratio, stage 4 disease

## Abstract

**Simple Summary:**

This study examined the outcomes of patients with gallbladder cancer over 20 years in Saskatchewan, Canada, focusing on geographic, demographic, and clinical factors. A total of 331 patients diagnosed between 2000 and 2019 were included. The majority (64%) had advanced stage 4 disease, with a significant proportion (66%) being rural residents. Key factors associated with poorer overall survival included stage 4 disease, lack of surgery, older age, a higher neutrophil-to-lymphocyte ratio, and no referral to a regional cancer center. Among early-stage patients, stage III disease and urban residence were associated with worse disease-free survival. The findings highlight the late-stage diagnosis and referral challenges impacting the outcomes of gallbladder cancer.

**Abstract:**

Background: Gallbladder cancer (GBC) is an uncommon cancer. This study aimed to determine the outcomes of GBC in relation to geographic, demographic, and clinical factors in a Canadian province from 2000 to 2019. Methods: This population-based retrospective cohort study included all patients diagnosed with gallbladder cancer (GBC) in Saskatchewan, Canada, from 2000 to 2019. Cox proportional multivariate regression analysis was conducted to identify factors associated with poorer outcomes. Results: In total, 331 patients with a median age of 74 years and male–female ratio of 1:2 were identified. Of these patients, 305 (92%) had a pathological diagnosis of GBC. Among patients with documented staging data, 64% had stage IV disease. A total of 217 (66%) patients were rural residents, and 149 (45%) were referred to a cancer center. The multivariate analysis for patients with stage I–III GBC showed that stage III disease [hazard ratio (HR), 2.63; 95% confidence interval (CI), 1.09–6.34)] and urban residence (HR, 2.20; 95% CI, 1.1–4.39) were correlated with inferior disease-free survival. For all patients, stage IV disease (HR, 3.02; 95% CI, 1.85–4.94), no referral to a cancer center (HR, 2.64; 95% CI, 1.51–4.62), lack of surgery (HR, 1.63; 95% CI, 1.03–2.57), a neutrophil–lymphocyte ratio of >3.2 (HR, 1.57; 1.05–2.36), and age of ≥70 years (HR, 1.51; 95% CI, 1.04–2.19) were correlated with inferior overall survival. Conclusions: In this real-world context, the majority of patients with GBC were diagnosed at a late stage and were not referred to a cancer center. For those with early-stage GBC, living in an urban area and having stage III disease were linked to worse outcomes. Across all stages of GBC, stage IV disease, older age, absence of surgery, lack of referral to a cancer center, and a high neutrophil-to-lymphocyte ratio were associated with poorer survival.

## 1. Introduction

Gallbladder cancer (GBC) is a rare and highly fatal gastrointestinal cancer. Although its incidence varies geographically, common clinical, epidemiological, and biological variables contribute to a universally poor prognosis for GBC [1]. The likelihood of favorable outcomes largely depends on early diagnosis and surgical resection. Notably, GBC mortality rates mirror the incidence rates across different regions [1]. The absence of effective screening tests, coupled with an aggressive tumor biology and delayed clinical presentation, often results in advanced-stage diagnosis [2]. In fact, only about 20% of patients with GBC are diagnosed at an early stage, which is among the lowest rates for all cancers [3]. The surgical resection of the gallbladder and adjacent structures is the primary curative treatment [4]. However, at the time of diagnosis, only about 10% to 20% of patients are suitable candidates for surgery with curative intent [2]. The complex anatomy of the portobiliary tract further complicates successful complete resection. Moreover, the postoperative recurrence rates for GBC are estimated to be as high as 65% [5].

Most cases of GBC are incidentally diagnosed during the pathological examination of the gallbladder following cholecystectomy [6]. Despite the established connection between gallstones and GBC, clinical assessment, serological tests, and imaging scans do not provide a reliable means of diagnosing the disease at an early stage [2,7]. The presence of cholecystopathies, such as porcelain gallbladder, may mask the presence of GBC, further complicating diagnosis [8]. This diagnostic challenge may be further exacerbated in rural areas, where reduced access to healthcare can delay diagnosis, leading to more advanced-stage disease at presentation. Rural residents with cancer may face poorer outcomes than their urban counterparts because reduced access to timely healthcare services may lead to delayed diagnosis and presentation at a later stage of the disease [9,10]. The rural–urban disparity in cancer outcomes is not homogenous across all cancer types. For instance, colorectal cancer and breast cancer tend to have significant rural–urban disparities in outcomes, whereas pancreatic cancer has minimal disparities [10,11,12,13]. However, rural–urban disparities often do not occur in isolation. Other epidemiological factors, such as race and education, contribute to the asymmetrical trends seen in rural–urban cancer outcomes [13,14].

The impact of urban or rural residence on the outcomes of GBC in North America has not been well investigated. Studies from other regions have shown conflicting results; some have revealed that rural residence may be associated with a higher incidence and worse outcomes of GBC, whereas others have shown an inverse relationship between urban residence and the prognosis of GBC [15,16]. The present study was performed to determine the prognostic significance of residence in combination with other factors in patients diagnosed with GBC in a Canadian province over a 20-year span in a real-world setting. Real-world evidence plays a significant role in understanding cancer outcomes because it provides insights from diverse patient populations in day-to-day clinical practice, reflecting the complexities of patient demographics, underlying comorbid illnesses, functional status, socioeconomic and contextual factors, and treatment patterns [17].

## 2. Methods

This retrospective cohort study, based on population data, included patients diagnosed with GBC in Saskatchewan, Canada, between January 2000 and December 2019. Proportionally, Saskatchewan has one of the largest rural populations in Canada. The patients were identified using the Saskatchewan Cancer Registry. They were classified as urban residents according to the Canadian 2011 census criteria [18]. The province has two urban comprehensive Cancer Centers, each covering half of the province. There are 16 rural Community Oncology Centers available for systemic therapy; however, they do not have an onsite cancer specialist for consultation.

The survival of the entire cohort and its subgroups was estimated using the Kaplan–Meier method, with survival distributions across different groups compared using the log-rank test. Disease-free survival (DFS) was calculated from the time of surgery to recurrence of the cancer, the development of a new primary invasive cancer, or death. Overall survival (OS) was calculated from the date of diagnosis to the date of death. GBC was clinically diagnosed in the presence of a gallbladder mass with metastatic lesions with an inconclusive or no tissue diagnosis. Major comorbid illnesses were defined as the presence of one or more concurrent conditions, including coronary artery disease, diabetes mellitus, chronic renal insufficiency, chronic obstructive lung disease, and other conditions such as uncontrolled hypertension, peripheral vascular disease, stroke or transient ischemic attack, interstitial lung disease, congestive heart failure, and cardiac arrhythmia, among others. Cox proportional multivariate regression analyses were conducted for DFS for stage I–III GBC and for OS for all stages of GBC. The hazard ratio (HR) and 95% confidence interval (CI) were calculated. The following variables were examined to determine their correlation with the risk of recurrence and mortality: age (<70 vs. ≥70 years), sex, WHO performance status (0 or 1 vs. >1), presence or absence of major comorbid illness, smoking history, residence (urban vs. rural), abnormal body mass index (<18.5 or ≥25 or 18.5 to <25), time period (2000–2009 vs. 2010–2019), surgery, referral to the cancer center, chemotherapy, radiation therapy, serum creatinine (≥100 vs. <100 mm/L), albumin (≥35 vs. <35 g/L), hemoglobin (≥120 vs. <120 g/L), alkaline phosphatase (≥120 vs. <120 U/L), elevated bilirubin (>22 µmol/L), anemia (hemoglobin <120 g/L), elevated aspartate aminotransferase (>45 U/L), median neutrophil–lymphocyte ratio (NLR), and stage of disease (stage I–III vs. stage IV). For DFS, the T status (T3 or 4 vs. T1 or 2), N status (node positive vs. negative), positive resection margin, and tumor grade (III vs. I or II) were also used in multivariate modeling. Patients with missing information were excluded from the multivariate analysis. In addition, to address the concern of multicollinearity between disease stage and the T and N status, these variables were removed from the model. The significance level of the *p* value was set at 0.05. All variables with a *p*-value of <0.20 in the univariate analysis were fitted in a multivariate model. A likelihood ratio test and *t*-test were employed to assess whether the inclusion of independent variables of interest added significantly the survival prediction in the model. A two-sided *p*-value of <0.05 was considered statistically significant. Statistical analysis was performed using SPSS version 29.0 (IBM Corp., Armonk, NY, USA).

## 3. Results

Overall, 331 patients with GBC were identified during the study period. Their median age at diagnosis was 74 years (range, 28–97 years), and the male–female ratio was 1:2. Of all patients, 49% were diagnosed prior to the year 2010. Among all patients, 305 (92%) had a pathological diagnosis of GBC, and 27 (8%) had a clinical diagnosis of GBC either without biopsy or with inconclusive pathology results (Table 1). Among patients with a pathological diagnosis, 80% had adenocarcinoma. Of 173 patients with documented symptoms, >90% were symptomatic at the time of diagnosis. The mean time from the onset of symptoms to diagnosis was 9.3 ± 9.0 months. Abdominal pain was experienced by 83% of patients, followed by jaundice (29%), weight loss of >5 kg (29%), and nausea or vomiting (27%). Other symptoms with a frequency of <10% included profound fatigue, diarrhea, melena, pruritus, and development of ascites. Of 232 patients with documented staging data, 64% had stage IV disease at the time of diagnosis. The remaining 28% of patients with undocumented staging data were suspected to have locally advanced inoperable and or metastatic disease. Of all patients, 149 (45%) underwent an oncology consultation at a cancer center.

Overall, 114 (34%) patients were urban residents and 217 (66%) were rural residents. Significant differences in smoking history, the pathological diagnosis of cancer, serum albumin concentration, and platelet count were noted between the two groups (Table 1). Rural residents had a significantly higher prevalence of a smoking history, while urban residents were more likely to have a confirmed pathological diagnosis, low serum albumin levels, and elevated platelet count. No significant differences in disease stage were noted between urban and rural patients or time period except for the fact that patients diagnosed during 2010–2019 had a significantly higher rate of stage III disease than patients diagnosed during 2000–2009 (Table 2). The difference in the mean time from onset of symptoms to diagnosis between urban and rural residents was not statistically significant (8.3 ± 7.2 vs. 10.1 ± 10.0 months; *p* = 0.18).

Of the 217 rural patients, 44% were seen at a cancer center, whereas 47% of urban patients were assessed by an oncologist (*p* = 0.5). The median age of patients not seen at a regional cancer center was 77 years [interquartile range (IQR), 68–85], while that of patients seen at a cancer center was 70 years (IQR, 63–78) (*p* < 0.001). Among the 182 patients who were not seen at a cancer center, 91% had advanced disease, in contrast to 62% of those who were seen at a cancer center (*p* < 0.001). Likewise, a pathological diagnosis was absent in only 1% of patients who were seen at a cancer center compared to 13% who were not seen at a cancer center (*p* < 0.001). Among 73 patients with definite stage I to III disease, no differences were noted in stage, age, performance status, comorbid illness, T and N status, NLR, or the use of adjuvant chemotherapy and radiation between urban and rural patients. Among patients with stage I to III disease, the median age of urban patients was 76 years (IQR, 65–84), and that of rural patients was 69 years (62–81) (*p* = 0.12). Among patients with early-stage disease, 27% of rural patients and 46% of urban patients had stage III disease (*p* = 0.12). Surgery was performed on 64 (88%) patients with stage I-III disease, and 41% of them had a radical surgery. Among them, 79% of urban patients underwent surgery compared to 93% of rural patients (*p* = 0.14). Of the 73 patients with early-stage disease, 16 (22%) were not referred to a cancer center. Among the 57 patients who were referred to a cancer center, 22 (39%) received adjuvant chemotherapy, and 10 (18%) received adjuvant radiation therapy. The most common adjuvant chemotherapy regimens were gemcitabine alone (23%) or in combination with cisplatin (23%), followed by single-agent capecitabine (18%). Of the 73 patients with early-stage disease, 27 (37%) experienced a relapse, with 41% relapsing in the liver, 22% in the peritoneum, and 11% each in the lungs and lymph nodes. Among those who relapsed, 22% underwent surgery, and 44% received palliative chemotherapy for recurrent disease. Among patients with advanced disease, 92 were seen at a cancer center, and 50 (54%) of them received palliative chemotherapy. The others were not candidates or declined treatment.

### 3.1. Survival

The median follow-up period for the entire cohort was 9 months, with a total follow-up duration after the time of first patient to the data cutoff date of 164 months. The median OS duration of all patients was 7 months (95% CI, 5.3–8.7). The 5-year estimated OS rate of the study cohort was 16%. Survival varied significantly by stage, with a 5-year OS rate of 91% for stage 0 disease and 1% for stage IV disease (Figure 1). The 5-year disease-free survival of stage I was 50%; stage II, 47%; and stage III, 12%. The 5-year disease-free survival of urban patients was 25% vs. 42% of rural patients (0.016) (Figure 2).

The median OS duration of patients with early-stage (I–III) disease was 20 months (95% CI, 10.9–29.1), whereas that of patients with stage IV disease was 4.0 months (95% CI, 3.0–5.0) (*p* < 0.001). Patients who were not seen for an oncology consultation had a median OS of 3 months (95% CI, 1.98–4.0), whereas those seen at a cancer center had a median OS of 13 months (95% CI, 9.9–16.1) (*p* < 0.001).

### 3.2. Cox Proportional Multivariate Analysis

The multivariate analysis of patients with stage I–III GBC showed that stage III disease (HR, 2.63; 95% CI, 1.09–6.34) and urban residence (HR, 2.20; 95% CI, 1.1–4.39) were correlated with a high risk of recurrence (Table 3). For all patients, stage IV disease (HR, 3.02; 95% CI, 1.85–4.94), no consultation at a cancer center (HR, 2.64; 95% CI, 1.51–4.62), lack of surgery (HR, 1.63; 95% CI, 1.03–2.57), a median NLR of >3.2 (HR, 1.57; 95% CI, 1.05–2.36), and age of ≥70 years (HR, 1.51; 95% CI, 1.04–2.19) were correlated with inferior overall survival. No treatment with chemotherapy and radiotherapy carried a non-significant HR of 1.18 (95% CI, 0.87–1.61; *p* = 0.27) and 1.28 (95% CI, 0.75–2.13; *p* = 0.37), respectively (Table 4).

## 4. Discussion

The present study showed that rural residence was not associated with inferior outcomes. Accessing standard-of-care treatment for a complex and relatively uncommon cancer such as GBC, which requires a multidisciplinary approach, may pose challenges for rural residents. Urban residents have been considered to have a logistical advantage in accessing such services. However, the present study showed an inverse relationship in relation to residence in patients with stage I–III GBC, with urban patients having inferior DFS compared with their rural counterparts. Notably, there were no significant differences in the stage at diagnosis between urban and rural patients. Similarly, there were no significant differences in age, comorbid illness, body mass index, or performance status between the two groups. However, rural patients had a significantly higher smoking rate compared to their urban counterparts. Urban patients with stage I–III disease tended to be older and have a higher incidence of stage III disease. As expected, stage III disease was also associated with an elevated risk of recurrence in patients with non-metastatic invasive GBC.

Other research groups have shown an inverse relationship between urban residence and the prognosis of GBC [15,16]. For example, a study of South Asian patients showed that urban residence was associated with a 56% higher mortality rate in patients with GBC [16]. There may be factors related to urban residence that contribute to inferior DFS, such as socioeconomic factors, environmental factors inherent to urban living, health literacy, and treatment adherence. While individual economic measures are not available in this study, a cursory assessment does not appear to associate GBC outcomes with income, at least on a macro level. Of Saskatchewan’s 51 postal code Forward Sortation Areas (FSA) with sufficient data, 28 are categorized as urban and 23 are categorized as rural [19]. As of 2019, 20 urban FSAs had a tax-return reported income above the provincial average, with these FSAs accounting for 57.5% of the urban population [19]. Conversely, only 10 out of 23 rural FSAs had a tax-return reported income above the provincial average, accounting for 31.2% of the rural population [19]. While this crude analysis can be confounded by many factors, such as individual income inequality and the prevalence of tax returns filed in rural areas, it does suggest that urban populations in Saskatchewan are, at the very least, not economically disadvantaged relative to rural populations. Nevertheless, there is a need for epidemiological studies that exclusively examine the relationship between socio-economic status and GBC outcomes.

For all stages of GBC, older age, lack of surgery, lack of consultation at a cancer center, and a high median NLR were correlated with inferior OS. According to previous studies, more than two-thirds of patients with GBC are diagnosed at surgery or postoperatively [20]. A lack of preoperative clinical suspicion of GBC combined with the lack of a specific diagnostic test commonly results in an advanced presentation, and this outcome is not affected by rural or urban residence [2]. In the present study, the mean time of diagnosis from the onset of symptoms in patients with documented symptoms was approximately 8 to 10 months. About two-thirds of patients in both urban and rural populations were diagnosed at stage IV disease, a rate in line with comparable studies and significantly higher than that of malignancies at other sites, including gastrointestinal cancers [21,22,23,24,25,26]. Therapeutic options in the latter stages of GBC are limited and, until very recently, had remained unchanged in their efficacy even across decades, as evidenced by the lack of a survival benefit by decade. The lack of association between receipt of chemotherapy and OS in the present study underscores this fact. Cumulatively, this suggests that the current standard-of-care for late-stage GBC is not appropriately targeted to the oncogenetic changes, which occur in advanced GBC [27].

In this study, lack of consultation at a regional cancer center was independently correlated with inferior OS irrespective of age, stage, or other clinical variables. Only 45% of patients were seen at a cancer center for systemic and/or radiation therapy. The referral of patients with suspected or diagnosed cancer to a cancer center can be influenced by numerous factors, including geographic location, healthcare provider preferences, patient preferences and beliefs, socioeconomic status, insurance coverage, the availability of specialized services, therapeutic nihilism, and mistrust in the healthcare system [27,28,29,30]. In this study, patients who were not seen at a cancer center were significantly older and more likely to have advanced disease. This finding is not isolated to either the present study or to gallbladder disease; elderly patients with breast, colorectal, and testicular cancer have also been shown to be significantly more likely to experience treatment delay [31,32,33]. Indeed, most GBC-related deaths in this study occurred in patients aged >75 years, complicating physician perspectives and patient preferences when it comes to referring elderly patients to a cancer center. Referral to a cancer center does not guarantee the receipt of adjuvant or palliative therapy. Many patients with advanced cancer seen at cancer centers may not receive systemic therapy. Several factors influence the use of chemotherapy, including patient preferences, clinical conditions, and socioeconomic support [34]. Achieving equity in outcomes for GBC patients in both urban and rural areas should not be hindered by the limited availability of effective therapies. The timely referral of GBC patients to a cancer center is crucial to improving outcomes by better aligning therapy delivery with the patient’s goals and preferences.

Systemic therapy provides modest survival and quality-of-life benefits in patients with stage IV GBC [35,36,37]. There is evidence of a survival benefit of adjuvant chemotherapy in patients with early-stage biliary tract cancer and GBC [37]. Furthermore, immunotherapy and targeted therapy have arguably shown the greatest promise in improving survival in patients with advanced GBC and bile duct cancer [38]. As of April 2024, 16 of 22 active clinical trials indexed by the National Cancer Institute recruiting patients with GBC involved targeted therapies [39]. Purposeful action should be taken to ensure that rural populations receive equitable access to clinical trials that offer such treatments. This is not only an ethical imperative; it is also necessary to increase the generalizability of trial results because study populations often do not match the composition of targeted populations [40]. However, logistical and financial barriers in conjunction with an indifferent regulatory culture presently discourage rural recruitment to clinical trials, potentially exacerbating future urban–rural inequities in GBC outcomes [40,41]. Our results show that while consultation at a cancer center was independently associated with better survival, the lack of chemotherapy in patients at all stages did not correlate with survival. This paradox might be explained by the heterogeneity of the patient population and the long study duration. Early studies suggested that chemotherapy provides modest benefits only in physically fit patients. Referral to a cancer center likely offers more than just chemotherapy, including multidisciplinary care, advanced diagnostics, and supportive care. These comprehensive services may improve overall outcomes, regardless of chemotherapy, particularly when tumor biology is the primary determinant of survival. Further research is needed to clarify the specific pathways through which cancer center referrals influence patient outcomes.

Patients with a high median NLR had inferior OS. The NLR is a biomarker indicative of inflammation and has been associated with inferior outcomes in various malignancies [42,43]. Mei et al. examined 66 studies involving 24,536 individuals and found that an elevated pretreatment NLR was associated with worse outcomes [42]. Several other studies have similarly shown that an elevated NLR is correlated with the inferior survival of patients with GBC [42,43]. Mady et al. explored the prognostic significance of the NLR using a cutoff of 5 in 231 patients with metastatic GBC [44]. They found that patients with an NLR of ≥5 and <5 had a median OS of 3.6 and 8.7 months, respectively. Using a cutoff of 2.61, Zhang et al. demonstrated that an elevated NLR was associated with inferior outcomes in patients with stage III and IV disease [45]. Chronic cholecystitis and gallbladder stones, commonly associated with GBC, may influence the NLR [46].

It is important to highlight some limitations of this study. In addition to the retrospective nature of the study, almost all patients were treated before access to targeted therapy and immunotherapy, both of which have improved the outcomes of patients with advanced biliary tract cancer and GBC. About 30% of patients were excluded from multivariate analysis due to missing variables. Furthermore, throughout most of the study period, level 1 evidence of the benefit of adjuvant chemotherapy was lacking. Additionally, we had no information on the patients’ socioeconomic status, education, and occupational status, which may have influenced the outcomes of the patients. Nevertheless, a key strength is that this is one of the largest studies of GBC in which individual patient data were collected, and the entire population of a Canadian province was evaluated over a span of 20 years. Despite its limitations, this study adds important information to the currently existing real-world evidence on GBC outcomes.

## 5. Conclusions

In summary, this retrospective population-based cohort study did not show a relationship between rural residence and negative outcomes; rather, urban patients with early-stage GBC had high recurrence rates. Most patients were diagnosed at a late stage of the disease, with the mean time to diagnosis being >6 months. In patients with advanced stage IV GBC, inferior OS was correlated with a lack of surgery, older age, a high NLR, and a lack of oncology consultation. Future prospective and qualitative studies are warranted to further explore the importance of contextual and other sociodemographic factors in the outcome of patients with GBC.

## Figures and Tables

**Figure 1 cancers-16-02932-f001:**
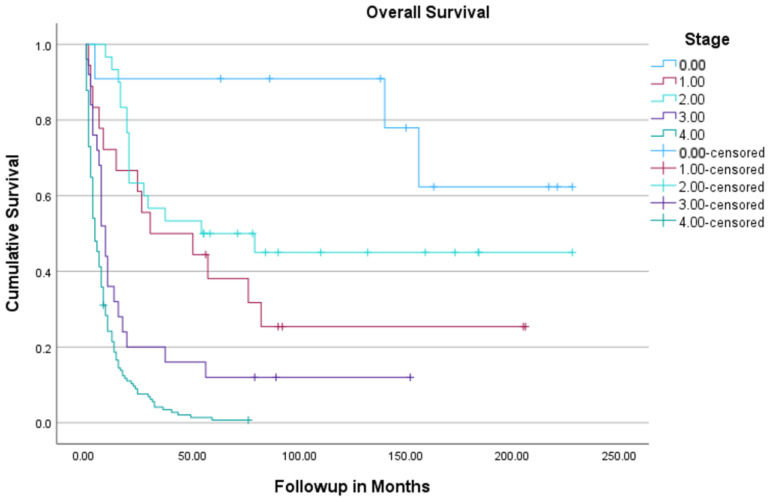
Kaplan–Meier overall survival of all patients with GBC based on stage of the disease.

**Figure 2 cancers-16-02932-f002:**
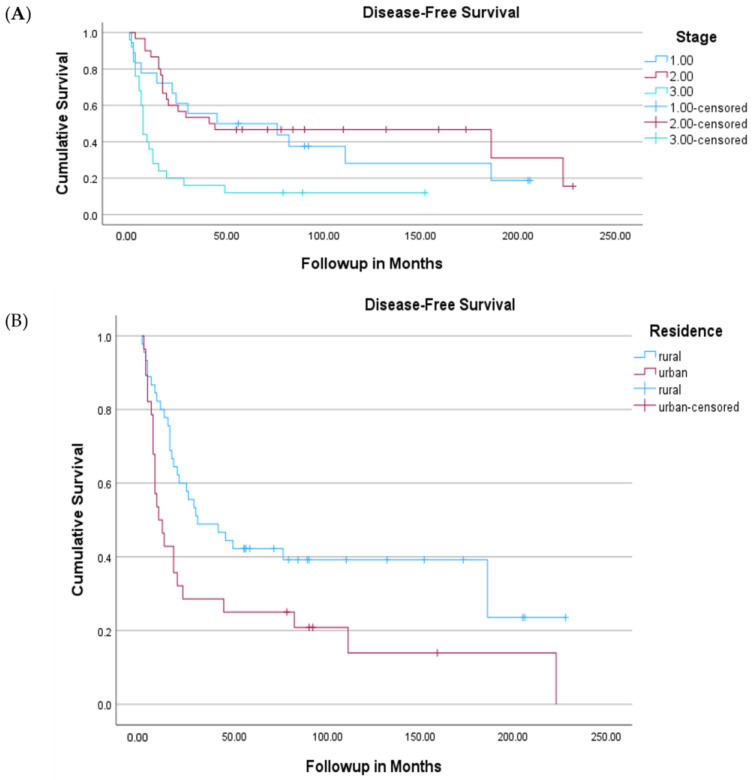
Kaplan–Meier Disease-free survival curve of patients with stage I to III disease based on stage (**A**,**B**) residence. Five-year DFS of stage 1 was 50%; stage 2, 47%; and stage 3, 12%. Five-year DFS of urban patients was 25% vs. 42% of rural patients (0.016).

**Table 1 cancers-16-02932-t001:** Baseline characteristics of patients with gallbladder cancer based on their residence.

Variables	All N = 331 %	Urban N = 114 %	Rural N = 217 %	*p* Values
Median age in years	74 (range: 28–97)	75 (36–93)	74 (28–97)	0.9
Men	33	32	34	0.8
Mean BMI (standard deviation)	28.1 ± 6.0	27.4 ± 6.1	28.7 ± 6.1	0.20
Comorbid illness ^a^	46	49	44	0.54
Smoking history ^b^	45	30	54	0.004
Diagnosed before year 2010	49	43	52	0.13
WHO performance status > 1 ^c^	40	44	37	0.44
Pathological diagnosis	92	97	90	0.03
Stage ^d^				
0	5	4	5	0.75
I	8	9	7	0.79
II	13	10	15	0.31
III	11	16	8	0.07
IV	64	62	65	0.77
Onset of symptoms to diagnosis in months (mean) ^e^	9.3 ± 9.0	8.3 ± 7.2	10.1 ± 10.0	0.18
Symptom Frequency				
Pain	83	85	83	1.0
Weight loss	29	32	27	0.4
Nausea/Vomiting	27	34	21	0.08
Jaundice	29	27	30	0.73
Oncology consult	45	47	44	0.56
Mean baseline laboratory values				
Mean creatinine	87 ± 57	78 ± 27	92 ± 68	0.18
Mean BUN	6.3 ± 4.7	6.2 ± 3.7	6.4 ± 5.2	0.77
Mean albumin	33 ± 7	31 ± 7	34 ± 7	0.05
Mean bilirubin	25.6 ± 58	20.3 ± 31	28.7 ± 69	0.22
Mean alkaline phosphatase	253 ± 360	233 ± 251	264 ± 413	0.64
Mean AST	72 ± 126	52 ± 86	83 ± 143	0.28
Mean ALT	59 ± 113	54 ± 87	62 ± 122	0.71
Mean WBC	9.8 ± 5.3	9.4 ± 4.0	9.9 ± 5.9	0.60
Mean hemoglobin	122 ± 19	122 ± 20	122 ± 18	0.97
Mean platelets	312 ± 122	341 ± 107	296 ± 127	0.04
Mean NLR	5.2 ± 4.8	5.6 ± 5.0	5.0 ± 4.6	0.50

± standard deviation; ALT: alanine transaminase; AST: aspartate aminotransferase; BUN: blood urea nitrogen; NLR: neutrophil–lymphocyte ratio. ^a^ Cormorbid illness was not documented in 156 patients; ^b^ smoking history was not documented in 177 patients; ^c^ WHO performance status was not reported in 143 patients; ^d^ staging information was not definite in 99 patients (these patients were suspected to have locally advanced and or metastatic disease); ^e^ symptoms were not documented in 158 patients.

**Table 2 cancers-16-02932-t002:** Stage information based on residence and time period of diagnosis.

Stage	Urban ^a^ N = 114 %	Rural ^b^ N = 217 %	*p* Value	Year 2000–2009 N = 162 %	Year 2010–2020 N = 168 %	*p* Value
0	4	5	1.0	8	3	0.15
I	9	7	0.52	8	8	1.0
II	10	15	0.31	15	12	0.41
III	16	8	0.13	4	15	0.005
IV	62	65	0.20	66	63	0.43

^a^ staging information was not definite in 32 patients (these patients were suspected to have locally advanced and or metastatic disease); ^b^ staging information was not definite in 67 patients (these patients were suspected to have locally advanced and or metastatic disease).

**Table 3 cancers-16-02932-t003:** Cox regression for DFS of patients with stage I to III gallbladder cancer.

Variables	Univariate Analysis		Multivariate Analysis	
	HR (95% CI)	*p* Value	HR (95% CI)	*p* Value
Age ≥ 70 years	1.52 (0.88–2.66)	0.13		
Men	1.13 (0.64–2.0)	0.67		
Urban residence	1.19 (1.11–3.31)	0.01	2.20 (1.1–4.39)	0.025
No comorbid illness	1.03 (0.55–1.94)	0.91		
Positive smoking history	1.30 (0.70–2.41)	0.40		
Abnormal body mass index	1. 15 (0.81–2.80)	0.20		
Year of diagnosis < 2010	1.14 (0.61–2.13)	0.67		
Stage III disease	3.1 (1.77–5.56)	<0.001	2.63 (1.09–6.34)	0.032
WHO performance status > 1	1.17 (0.56–2.44)	0.68		
No referral to a cancer center	1.01 (0.52–1.92)	0.98		
No surgery	7.30 (1.63–32.45)	0.009		
No chemotherapy	1.35 (0.72–2.55)	0.34		
No radiation	1.71 (0.80–3.65)	0.16		
T3 disease	2.75 (1.56–4.88)	<0.001		
Node-positive disease	1.14 (0.64–2.02)	0.65		
Grade 3 tumor	2.29 (1.31–4.0)	0.004		
Positive resection margin	2.06 (1.10–3.86)	0.024		
Year of diagnosis < 2010	1.14 (0.61–2.13)	0.67		
Albumin < 35 g/L	1.18 (0.66–2.09)	0.58		
Creatinine > 100 µm/L	1.03 (0.61–1.84)	0.82		
Total bilirubin > 22 µm/L	1.52 (0.89–2.62)	0.12		
Alkaline phosphatase > 120 IU/L	1.73 (0.99–3.04)	0.056		
AST > 45 U/L	1.96 (1.05–3.69)	0.04		
Hemoglobin < 120 g/L	1.26 (0.73–2.18)	0.40		
Platelet > 450 × 10^9^	1.29 (0.18–9.36)	0.80		
NLR > 3.2	1.22 (0.62–2.39)	0.56		

**Table 4 cancers-16-02932-t004:** Cox regression for OS all patients with stage 0 to IV disease.

Variables	Univariate Analysis		Multivariate Analysis	
	HR (95% CI)	*p* Value	HR (95% CI)	*p* Value
Age ≥ 70 years	1.52 (1.20–1.95)	<0.001	1.51 (1.04–2.19)	0.03
Men	1.17 (0.91–1.95)	0.20		
Urban residence	1.15 (0.90–1.47)	0.25		
Comorbid illness	1.35 (0.98–1.86)	0.07		
Positive smoking history	1.24 (0.88–1.74)	0.22		
Abnormal body mass index	1.15 (0.83–1.58)	0.42		
Year of diagnosis < 2010	1.12 (0.88–1.41)	0.36		
Stage IV disease	3.8 (2.71–5.41)	<0.001	3.02 (1.85–4.94)	<0.001
WHO performance status > 1	2.62 (1.89–3.61)	<0.001		
No surgery	3.50 (2.53–4.81)	<0.001	1.63 (1.03–2.57)	0.03
No referral to a cancer center	1.63 (1.29–2.06)	<0.001	2.64 (1.51–4.62)	<0.001
No chemotherapy	1.18 (0.87–1.61)	0.27		
No radiation	1.28 (0.75–2.13)	0.37		
Albumin < 35 g/L	1.79 (1.28–2.49)	<0.001		
Creatinine > 100 µm/L	1.31 (1.01–1.69)	0.04		
Total bilirubin > 22 µm/L	1.59 (1.22–2.06)	<0.001		
Alkaline phosphatase > 120 IU/L	2.03 (1.44–2.87)	<0.001		
AST > 45 U/L	1.91 (1.35–2.72)	<0.001		
Hemoglobin < 120 g/L	1.63 (1.23–2.17)	<0.001		
Platelet > 450 × 10^9^/L	1.50 (0.92–2.47)	0.10		
NLR > 3.2	1.30 (0.98–1.74)	0.07	1.57 (1.05–2.36)	0.03

## Data Availability

The data presented in this study are not publicly available. Data access will require approval from the University of Saskatchewan Biomedical Ethics Board and Data Access Committee of the Saskatchewan Cancer Agency.

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
