# Peer review of "Prognostic Factors in Patients Diagnosed with Gallbladder Cancer over a Period of 20 Years: A Cohort Study"

_cancers, 2024, doi:10.3390/cancers16172932_

Round 1

Reviewer 1 Report

Comments and Suggestions for Authors

Interesting retrospective article on the incidence of gallbladder cancer in rural and urbanized areas of Canada. The clinical history of this neoplasm is mentioned in the introduction. We agree that the neoplasm has an insidious onset, it is almost always an occasional finding after a simple cholecystectomy in patients who knew they were suffering from gallstone lithiasis. Other times there are occasional cholecystopathies such as porcelain gallbladder which hide a neoplasm and which are totally asymptomatic (doi.org/10.1007/s10353-021-00710-2 possibly to be cited in the bibliography). In other circumstances, patients appear jaundiced or with other complications such as perforation, empyema or cholecystitis which hide the neoplasm so that even with an emergency CT scan it is not possible to make a differential diagnosis between inflammation and tumor. We have nothing to add regarding the materials and methods, we can only state that the retrospective study takes into consideration a cross-section of the population which, due to age or other habits, presents classic pathologies of the Western world. The medical attitude can only be shared by adding that all patients, even those with a type of pT1A disease, must be sent for follow-up to the relevant oncologist. Fortunately, there are few cases whose diagnosis precludes surgical treatment. The discussion highlights that diagnostics and therapy do not vary between rural and urbanized populations and this is obviously indicative of a socially, economically and culturally advanced country. We hope that in a future world there will no longer be these differences and this paper must serve as an example. I would recommend recomposing the sentences: "Several variables influence the use of chemotherapy, including patient preference, clinical conditions, and contextual and socioeconomic support (31). Equity between urban and rural outcomes in patients affected by GBC should not depend on the relative scarcity of effective therapies offered in both rural and urban populations, timely referral of GBC patients to a cancer center is important to improve GBC outcomes through better alignment of therapy delivery. systemically with the patient's goals and preferences". This is to confirm what is stated in the paper. We agree on the therapeutic principles, adding only that, given that the majority of cholecystectomies are performed with laparoscopic access, it is suggested that its removal is always ensured with the endobag to avoid skin insemination. Good iconography accompanies the work, excellent English, good bibliography.

Reviewer 2 Report

Comments and Suggestions for Authors

This is a retrospective cohort analysis of a gallbladder cancer occurring in a single Canadian province over 20 years.  Comprehensive baseline registry clinicopathological data were available, as well as DFS and OS.  My comments/questions are as follows:

11.       “Contextual” is used several times throughout the manuscript, however, the only non-demographic and non-clinical factor described was location of residence so perhaps “geographic” would be more appropriate?

22.       Were regional cancer centres considered to be distinct from urban/metropolitan cancer centres?

33.       Lines 159/160 – are oncologists only available to urban areas?  ie not available in regional cancer centres?

44.       How many patients received chemotherapy, either adjuvant or palliative?

55.       Table 3 – the UVA values for “Year <2010” and “No referral to cancer centre” are missing

66.       Table 3 c/w table 4 – is “cancer centre” the same as “regional cancer centre?”

77.       Table 4 – it would be helpful to include rural residence in the UVA to demonstrate that this is/is not associated with OS.  Particularly as a key conclusion is that rural location does not adversely impact on survival outcomes.

88.       Table 4 – I am still confused about what a “regional cancer centre” is.  Do only rural patients have access, or do urban patients also have access to these institutions?

99.       The discussion highlights the significant associations found in the MVAs, however, only attempts to explain them in generic terms.  Particularly in view of the somewhat paradoxical finding that stage I-III urban patients do worse in terms of DFS, it would be important for instance to try to identify confounding factors which may explain this.  An obvious one would be socio-economic status – which may not be available for individual patients but should be available, for instance, at a postcode or similar level

110.   The similarly paradoxical lack of association of chemotherapy receipt with OS counterpointed by the clear association of lack of referral to a regional cancer centre with OS also should be explained.  (likely the disease biology is a stronger factor than treatment)

111.   Line 300 – “negative” should be omitted
